# DeTeCtive: Detecting AI-generated Text via Multi-Level Contrastive Learning

Xun Guo[1]*   Shan Zhang[2]*   Yongxin He[2]*   Ting Zhang[2]
Wanquan Feng[1]   Haibin Huang[1]†   Chongyang Ma[1]
[1]ByteDance     [2]University of Chinese Academy of Sciences

## Abstract

Current techniques for detecting AI-generated text are largely confined to manual feature crafting and supervised binary classification paradigms. These methodologies typically lead to performance bottlenecks and unsatisfactory generalizability. Consequently, these methods are often inapplicable for out-of-distribution (OOD) data and newly emerged large language models (LLMs). In this paper, we revisit the task of AI-generated text detection. We argue that the key to accomplishing this task lies in distinguishing writing styles of different authors, rather than simply classifying the text into human-written or AI-generated text. To this end, we propose **DeTeCtive**, a multi-task auxiliary, multi-level contrastive learning framework. DeTeCtive is designed to facilitate the learning of distinct writing styles, combined with a dense information retrieval pipeline for AI-generated text detection. Our method is compatible with a range of text encoders. Extensive experiments demonstrate that our method enhances the ability of various text encoders in detecting AI-generated text across multiple benchmarks and achieves state-of-the-art results. Notably, in OOD zero-shot evaluation, our method outperforms existing approaches by a large margin. Moreover, we find our method boasts a Training-Free Incremental Adaptation (TFIA) capability towards OOD data, further enhancing its efficacy in OOD detection scenarios. We will open-source our code and models in hopes that our work will spark new thoughts in the field of AI-generated text detection, ensuring safe application of LLMs and enhancing compliance.[3]

## 1 Introduction

Recently, the field of large language models (LLMs) [6, 12, 60, 69] has witnessed swift advancements, bringing great convenience to both professional settings and daily life. However, the widespread use of AI-generated text also poses threats to global information security, manifesting in the propagation of disinformation, misinformation, and content that can incite harmful or destructive behaviors [16]. Hence, the detection of AI-generated text has ascended as a task of vital importance.

On the other hand, with the advancement of LLMs, the task of AI-generated text detection has elevated into an escalating challenge. Early methods, such as watermarking methods [23, 32] and statistical-based methods [63, 47] encountered performance bottlenecks due to their reliance on manually hand-crafted forms. Moreover, the inherent inability to swiftly adapt to newly emerged LLMs further restricts their effectiveness. In stark contrast, recent training-based methods [11, 27, 24] have showcased notable improvements in performance. However, they remain constrained by the necessity of precisely paired training data and exhibit unsatisfactory generalization in out-of-distribution (OOD) detection scenarios due to the fixed binary classification formulation.

---

*main contributor

†Corresponding author: `jackiehuanghaibin@gmail.com`

[3]Our code is available at `https://github.com/heyongxin233/DeTeCtive`

38th Conference on Neural Information Processing Systems (NeurIPS 2024).

In this paper, to overcome these challenges, we revisit AI-generated text detection and approach the problem from a fresh perspective. Individual authors invariably exhibit unique writing styles, collectively constituting a vast feature space of writing styles. Our key insight is that an LLM can be viewed as a specific author, with the text it generates conforming consistently to its unique style. In line with this key observation, we propose to reformulate AI-generated text detection as a task of distinguishing diverse writing styles within the feature space, rather than merely treating it as a binary classification problem between human-written and AI-generated. This reformulation presents a fresh perspective from which to approach the detection of AI-generated text.

While distinguishing writing styles within a vast feature space may seem more challenging than binary classification, we can take advantage of mature techniques within the field of Natural Language Processing (NLP). Specifically, contrastive learning [9, 25, 21] employs a self-supervised approach to identify similarities and differences between positive and negative samples, thereby acquiring discriminative feature representations. These representations facilitate the differentiation of writing styles, enabling us to comprehend the characteristic patterns of different sources.

Specifically, we propose a general framework that combines a novel multi-level contrastive learning with multi-task learning tailored for AI-generated text detection. Our method enhances the writing-style encoding capabilities of various models, including but not limited to BERT-based [18] and T5-based [51] models. This framework is capable of calibrating the distances between samples sharing different degrees of relatedness, thereby encoding distinctive features of text generated by different authors (either humans or LLMs). During inference, we propose a pipeline anchored by dense information retrieval [58, 66]. Firstly, we pre-encode data drawn from the training dataset, extract features and store them within a feature database. Then, for any given query text, we simply calculate the similarity between its encoded feature and each feature vector nestled in the feature database. This measure is used to evaluate the degree of writing-style similarity. Finally, we employ the K-Nearest Neighbors (KNN) [15] algorithm for classification prediction.

Applying our method across multiple commonly-used datasets consistently improves performance with various text encoders compared to their zero-shot baselines, exceeding current solutions and establishing new state-of-the-art benchmarks on each individual dataset. Impressively, our method also demonstrates superior generalization capabilities when faced with OOD data emerging from domains or models that are not encountered during the training phase. Specifically, the Average Recall (AvgRec) metrics on the Unseen Models and Unseen Domains test sets from the Deepfake [39] dataset outperform existing state-of-the-art solutions by **5.58%** and **14.20%**, respectively.

Additionally, we introduce Training-Free Incremental Adaptation (TFIA), a novel and efficient scheme for boosting the generalization capability for OOD detection. Particularly, when confronted with a batch of OOD data, our goal is to enhance the model's adaptability to unseen domains using these data. The existing solutions either involve retraining the model or fine-tuning it on the new data. Contrastingly, under our framework, we discover that no further training is necessary. We simply encode these data using our previously trained model and incorporate them into the existing database to create an augmented database. Notably, within the aforementioned OOD detection scenarios, TFIA contributes to a further improvement in model performance: The AvgRec score witnesses an **additional** increase of **0.84%** on Unseen Models, and a noteworthy **7.03%** on Unseen Domains.

Extensive experiments across several datasets and models consistently demonstrate that our proposed method outperforms previous approaches, establishing new state-of-the-art performance. This superiority is maintained in both In-distribution and OOD detection scenarios. In summary, the contributions of our study are manifold, and can be enumerated as follows:

- We propose a novel end-to-end framework for AI-generated text detection, wherein we carefully devise a multi-task auxiliary, multi-level contrastive loss to learn fine-grained features for distinguishing various writing styles.

- We present Training-Free Incremental Adaptation (TFIA), a key feature of our method. Utilizing a modest amount of OOD data, TFIA enhances the model's adaptability to new domains without further training, offering significant advantages for practical applications.

- Our method achieves state-of-the-art performance on multiple datasets in both In-distribution and OOD detection scenarios, substantially surpassing existing methods.

- We validate the effectiveness of each component through a series of ablation studies and provide visualization results for further analysis. Furthermore, we perform detailed experiments on TFIA and provide an empirical analysis.

## 2  Related Work

**AI-generated text detection.**   Existing methods for AI-generated text detection generally fall into the following three categories: (i) *Watermarking methods*: watermarking methods, which include rule based [5, 30, 59] and deep learning based [17, 62] methods, involve embedding specific markers into AI-generated content, which can later be used to verify its source. The soft watermarking method [32] is an inference-time framework that involves grouping the vocabulary and decoding the next token preferentially. [23] proposes a method of adding watermarks by embedding backdoors triggered by special inputs into the model. UPV [41] is an unforgeable and publicly verifiable algorithm ensuring security against forgery and unauthorized detection attempts. (ii) *Statistical methods*: applying statistical metrics like entropy as thresholds to distinguish AI-generated text from human-written text. HowkGPT [63] identifies text origins by comparing perplexity scores of human-written and ChatGPT [6, 69] generated text. DetectGPT [47] utilizes the structural properties of the LLM's probability density for zero-shot detection of AI-generated text. Similarly, DetectLLM [56] employs normalized perturbation log-ranks for identification, exhibiting less sensitivity to perturbations. (iii) *Supervised learning methods*: GPT-Sentinel [11] incorporates a binary classifier into RoBERTa [43] and T5 [51], which are directly trained on specific datasets. RADAR [27] employs an adversarial learning approach. By continually iterating to improve the detector and generator (both of which are LLMs), RADAR performs well in detecting both original and paraphrased AI-generated text. [55] utilizes contrastive learning to learn style representations on human-written text and uses the learned representations to identify different sources in a few-shot manner. Building on SCL [24] framework, CoCo [42] incorporates coherency information into the text representation, enhancing the ability to detect AI-generated text under resource-constrained conditions.

**Contrastive learning for NLP.**   The success of MoCo [25] and SimCLR [9] in the field of Computer Vision through contrastive learning has prompted research efforts to explore its potential in the area of Natural Language Processing (NLP), resulting in the development of various strategies to enhance text encoding capabilities via contrastive learning. IS-BERT [78] employs the DIM [26] framework to learn text representations. The ArcCon loss [80] is proposed to further enhance the model's semantic discriminating ability. MixCSE [79] introduces an unsupervised method for text representation learning, which incorporates a mixed negative sample strategy to boost the model's ability to discriminate complex semantics. VaSCL [75] adopts a more general approach to procure hard negatives by defining an instance-level contrastive loss and integrating Gaussian noise, it effectively enhances the model's performance in an unsupervised manner. DCLR [82] addresses the anisotropic problem brought about by negative samples in unsupervised sentence representation learning by introducing noise-based negative samples and virtual adversarial training, thereby improving the uniformity of the representation space. SimCSE [21] proposes to predict the input sentence itself, utilizing standard dropout as noise in an unsupervised manner. They also introduce a method for categorizing positive and hard negative sample pairs, thereby improving the sentence representations.

## 3  Method

In this section, we provide a detailed description of the proposed method. We begin in Section 3.1 with a definition of AI-generated text detection and an overview of our proposed framework. In Section 3.2, we explore the design of the multi-task auxiliary multi-level contrastive learning, which are critical components of our framework. Finally, in Section 3.3, we introduce Training-Free Incremental Adaptation (TFIA), an efficient and effective strategy that leverages our method's generalization capability to handle out-of-distribution (OOD) data.

### 3.1  Framework Overview

In this work, we focus on the task of AI-generated text detection. Given a query text $x$ with $L$ tokens, $x = \{w_1, w_2, ..., w_L\}$, we aim to determine whether it is human-written or AI-generated. Existing

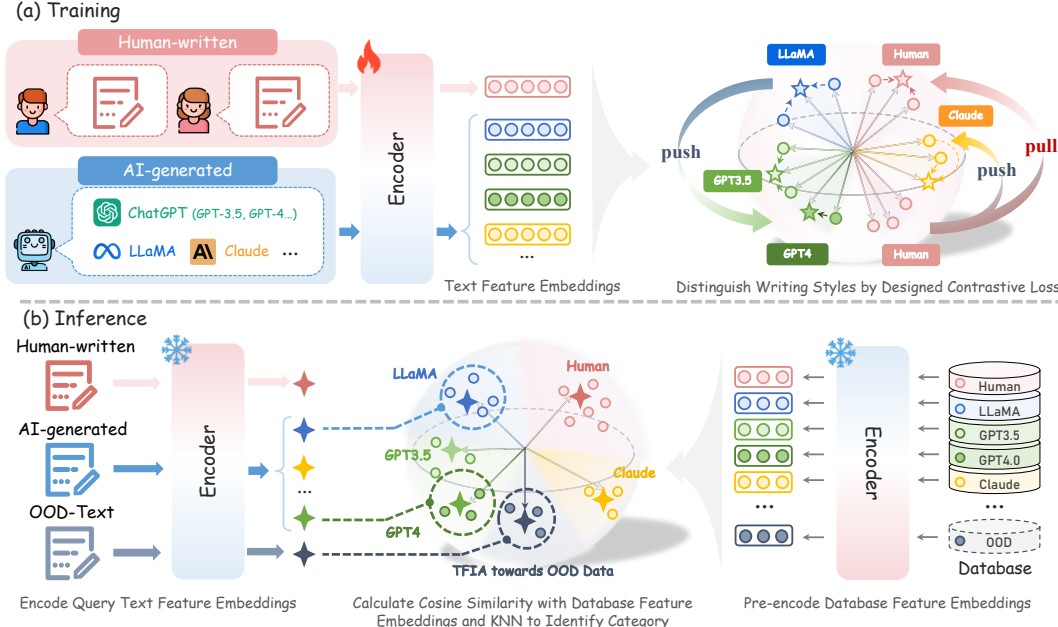

Figure 1: Overview of DeTeCtive. (a) Training. With our proposed multi-task auxiliary multi-level contrastive loss, the pre-trained text encoder is fine-tuned to distinguish various writing styles. (b) Inference. We employ a similarity query-based method for classification and incorporate Training-Free Incremental Adaptation (TFIA) for out-of-distribution (OOD) detection.

methods typically employ hand-crafted features [63, 47, 32] or adopt neural networks [11, 27, 24] to learn discriminative features between human-written and AI-generated text, treating them as two distinct categories. Ultimately, this task is reduced to a binary classification problem. While this formulation appears simple and straightforward, it neglects a vital factor. Analogous to how different novelists often demonstrate unique writing styles, it's critical to consider that different LLMs, due to variations in model architectures, training data and strategies, will inevitably infuse certain preferences and biases. Consequently, these variations induce stylistic differences. Therefore, categorizing all the texts generated by any LLM as the same category clearly overlooks these disparities.

To overcome this limitation, we present **DeTeCtive**, a general framework compatible with diverse text encoders, as shown in Figure 1. By leveraging a method that incorporates a novel multi-level contrastive learning with multi-task learning, DeTeCtive regulates the distance between samples of varying relations within the feature space, enabling the model to learn distinctive features. During inference, we adopt a dense information retrieval [66, 58] pipeline. The query text is classified by comparing its similarity with existing data entries in the database via the K-Nearest Neighbors (KNN) [15] algorithm.

## 3.2 Multi-task Auxiliary Multi-level Contrastive Learning

**Optimization objective and justification.** As discussed in Section 3.1, the distinctive writing styles attributed to different authors constitute a vast feature space. We perceive each LLM as an *individual author*. Consequently, AI-generated text detection evolves into a task of differentiating diverse writing styles within this feature space. Driven by this insight, it becomes critical to discern the similarities and discrepancies across varying writing styles. To effectuate this, we carefully devise a multi-task auxiliary, multi-level contrastive loss to facilitate the learning of fine-grained features.

Specifically, LLMs developed by the same company often demonstrate similar preferences and inherent biases, given the shared model designs, training strategies, and datasets utilized [60, 13, 76, 70]. Common techniques [81] like the unified auto-regressive modeling approach can also introduce some level of commonalities across company boundaries, though these may be less pronounced. Drawing parallels, the multi-level similarities among LLMs can be seen as familial kinship relations within an expansive family tree, distinguishing between those closely related and those more distant.

We aim to capture these kinship relations with a text encoder, allowing the encoder to capture the multi-level similarities and distinctions. Consequently, we expect the encoded features from different sources to reflect their relations within the high-dimensional feature space as follows:

$$\mathbb{E}_{x \sim P_i, y \sim P_j}[Sim(\Phi(x), \Phi(y))] > \mathbb{E}_{x \sim P_i, y \sim P_{j+1}}[Sim(\Phi(x), \Phi(y))], \forall 1 \leq i \leq j < 4, \quad (1)$$

where $Sim$ denotes the similarity measurement, $\Phi(\cdot)$ symbolizes the encoding function, and $P_1$ to $P_4$ signify different text distributions. Specifically, $P_1$ corresponds to the distribution generated by a particular LLM, $P_2$ to the distribution generated by LLMs developed by the same company, $P_3$ to the distribution generated by any LLM, and $P_4$ to the distribution of human-written text. This configuration aims to ensure that closeness in distribution corresponds to heightened similarity after encoding, encouraging the model to discern fine-grained multi-level relations.

**Multi-level contrastive learning.** According to the similarity constraints defined in Ineq. 1 above, when processing a data batch containing $N$ samples, for the $i_{th}$ sample $T_i$, we assign it with a label $x_i$. If the text is generated by an LLM, then $x_i = 0$, otherwise, $x_i = 1$. For those AI-generated text (i.e., $x_i = 0$), we further label the model series and the specific model with $y_i$ and $z_i$. Then, the encoding function $\Phi(\cdot)$ maps the text into a $d$-dimensional feature space $R^d$. For any two samples $T_i$ and $T_j$, we compute the cosine similarity between their encoded features through $Sim(\Phi(T_i), \Phi(T_j))$, and define this similarity metric as $S(i, j)$. For human-written text $T_i(x_i = 1)$, the similarity of its encoding with other human-written text encodings should be greater than the similarity with AI-generated ones, hence the following relationship should be satisfied:

$$S(i, j) > S(i, k), \forall x_j = 1, x_k = 0. \quad (2)$$

Similarly, for text $T_i(x_i = 0)$ generated by LLMs, Ineq. 1 suggests the existence of multi-level similarities and differences internally within LLMs, expressed as follows:

$$S(i, j) > S(i, l) > S(i, m) > S(i, n), \forall z_i = z_j, z_i \neq z_l, y_i = y_l, y_i \neq y_m, x_i = x_m, x_i \neq x_n. \quad (3)$$

In order to achieve the above optimization objectives, we propose a method to solve these constraints hierarchically. Specifically, for the first inequality in Ineq. 3, we consider the index $l, m, n$ that satisfies the conditions in the above constraints as a whole set, denoted as $k$, that is:

$$S(i, j) > S(i, k), \quad \forall z_i = z_j, z_i \neq z_k. \quad (4)$$

For the remaining inequalities, similar conditions are set to satisfy the constraints, culminating in:

$$\begin{cases} S(i,j) > S(i,k), & \forall x_i = 1, & x_i = x_j, & x_i \neq x_k \\ S(i,j) > S(i,k), & \forall x_i = 0, & z_i = z_j, & z_i \neq z_k \\ S(i,j) > S(i,k), & \forall x_i = 0, & z_i \neq z_j, & y_i = y_j, & y_i \neq y_k \\ S(i,j) > S(i,k), & \forall x_i = 0, & y_i \neq y_j, & x_i = x_j, & x_i \neq x_k. \end{cases} \quad (5)$$

To address the similarity constraints defined in Ineq. 5, we adopt a framework based on SimCLR [9] and propose a method for defining positive and negative sample pairs, from which we derive the corresponding contrastive learning loss. Unlike conventional contrastive losses, our positive sample is not a single instance, but a collection of positive samples meeting the conditions. We consider the positive sample similarity as the average value related to the entire set of positive samples from the current sample's perspective. The handling of negative samples echoes that of SimCLR, rendering the contrastive learning loss as demonstrated in Eq. 6, where $q$ signifies the current sample, $K^+$ is a set of positive samples, $K^-$ is a set of negative samples, $\tau$ indicates the temperature coefficient, and $N_{K^+}$ represents the size of the positive sample set.

$$\mathcal{L}_q = -\log \frac{\exp\left(\sum_{k \in K^+} \frac{S(q,k)}{\tau}/N_{K^+}\right)}{\exp\left(\sum_{k \in K^+} \frac{S(q,k)}{\tau}/N_{K^+}\right) + \sum_{k \in K^-} \exp\left(\frac{S(q,k)}{\tau}\right)}. \quad (6)$$

Different constraints correspond to varied positive and negative sample sets, and accordingly, multi-level contrastive losses are calculated. Following the definition in Ineq. 5, these loss are denoted as $\mathcal{L}_{q_i,1}, \mathcal{L}_{q_i,2}, \mathcal{L}_{q_i,3}, \mathcal{L}_{q_i,4}$, respectively. The overall multi-level contrastive loss $\mathcal{L}_{mcl}$ is as shown in Eq. 7, where $\delta, \alpha, \beta$, and $\gamma$ are coefficients used to adjust the weight between the multi-level relations. Take note that we designate $\delta$ as the coefficient balancing human-written and LLMs-generated, ensuring $\delta = \alpha + \beta + \gamma$, in an effort to maintain equilibrium, and we set $\alpha = \beta = \gamma = 1.0$.

$$\mathcal{L}_{mcl} = \sum_{i=1}^{N} x_i \cdot (\delta \cdot \mathcal{L}_{q_i,1}) + (1 - x_i) \cdot (\alpha \cdot \mathcal{L}_{q_i,2} + \beta \cdot \mathcal{L}_{q_i,3} + \gamma \cdot \mathcal{L}_{q_i,4}). \quad (7)$$

Through this carefully designed multi-level contrastive learning, we drive the model to learn fine-grained features of different sources. This strategy empowers the model to discern diverse writing styles, enhancing the accuracy and generalization of AI-generated text detection.

**Multi-task auxiliary learning.** Given that multi-task learning [7] enables the model to simultaneously learn multiple tasks online by sharing useful information between different tasks, it promotes the model to learn more generic and discriminative features, hence enhancing the model's generalization ability. Therefore, based on the aforementioned contrastive learning framework, we integrate an MLP classifier into the output layer of the encoder. This classifier performs a binary classification to determine whether a given query text was generated by human or LLM. We introduce a cross-entropy loss $\mathcal{L}_{ce}$ to optimize this classifier as follows:

$$\mathcal{L}_{ce} = -\frac{1}{N} \sum_{i=1}^{N} x_i \cdot log(p_i) + (1 - x_i) \cdot log(1 - p_i), \tag{8}$$

where $p_i$ is the probability of the $i_{th}$ sample $x_i$ being classified as human-written. Therefore, the overall multi-task auxiliary multi-level contrastive loss is defined as:

$$\mathcal{L}_{all} = \mathcal{L}_{mcl} + \mathcal{L}_{ce}. \tag{9}$$

### 3.3 Training-Free Incremental Adaptation

With the rapid advancement of LLMs and their proliferating applications, new models continually emerge, spanning an increasingly diverse range of domains. Existing AI-generated text detection solutions, which typically treat the task as a binary classification problem [11, 24], encounter difficulties in generalizing to new models and domains that yield out-of-distribution (OOD) data. When confronted with OOD data, these approaches commonly require retraining the model, a strategy that undeniably falls short of practicality in real-world applications. In light of this challenge, we propose a novel solution based on our existing framework — the Training-Free Incremental Adaptation (TFIA). This method allows our model to adapt to new domains or newly emerged LLMs without any further training. Specifically, When encountering OOD data not covered in the training set, we simply encode these data using our fine-tuned text encoder and incorporate the encoded features into the existing feature database $D_E$, forming an expanded feature database $D'_E$. During inference, replacing the original database $D_E$ with the expanded feature database $D'_E$ can enhance the performance of the model when dealing with OOD data. TFIA amplifies DeTeCtive's ability in identifying OOD sources, effectively leveraging the model's generalization capabilities. Through this mechanism, the DeTeCtive framework can adapt to OOD data without any retraining. We validate the effectiveness of TFIA through a series of experiments.

## 4 Experiments

In this section, we first introduce the utilized datasets, evaluation metrics, baseline methods, and implementation details in Section 4.1. We then present main experimental results and other applications in Section 4.2 and Section 4.3, followed by ablation studies and Training-Free Incremental Adaptation (TFIA) analysis in Section 4.4.

### 4.1 Experimental Setup

**Datasets.** In this study, we employ three widely-used and challenging datasets to evaluate our proposed method. The *Deepfake* [39] dataset includes text generated by 27 different LLMs and human-written content from multiple websites across 10 domains, encompassing 332K training and 57K test data. It also outlines six diverse testing scenarios, covering an array of settings from domain-specific to cross-domains, and out-of-distribution detection scenarios. The *M4* [68] dataset is a multi-domain, multi-model, and multi-language dataset encompassing data from 8 LLMs, 6 domains, and 9 languages. With machine text in its testing data paraphrased by OUTFOX [33], which introduces more complexity to the task. We perform experiments in both monolingual and multilingual settings, with the former containing 120K training and 34K testing data, and the latter comprising 157K training and 42K testing data. Finally, we make use of the *TuringBench* [61] dataset. TuringBench collects human-written text mainly from news titles and content, predominantly

politics-related. Incorporating data from 19 LLMs within a single domain, it forms a dataset of 112K training and 37K testing entries. For more detailed information, please refer to Appendix C.

**Evaluation metrics.** In line with existing works, we employ Average Recall (AvgRec) and the F1-score as our primary evaluation metrics. AvgRec, the average of recall for human-written (HumanRec) and AI-generated (MachineRec) text. Simple accuracy is inadequate for reflecting a model's performance on a minority class, especially in cases of data imbalance. The F1-score considers both the precision and recall of the model, evaluating overall model performance by computing the harmonic mean of these two. Together, these metrics present a comprehensive view of the effectiveness in detecting AI-generated text.

**Baseline methods.** In the experiment assessing the compatibility of our method to various text encoders, we use the zero-shot results of these pre-trained text encoders on the Cross-domains & Cross-models subset of the Deepfake dataset as the baseline. We then compare these results with the ones after fine-tuning with our method. In all subsequent experiments, for comparison analysis, we utilize the pre-trained SimCSE-RoBERTa [21] model as our text encoder. We conduct comparisons with several training-based methods across all three datasets. These incorporate methods which train classifiers upon RoBERTa [43] and Longformer [2] models, the T5-Sentinel [10] method that classifies using the output probability of the T5 [51] model, and the SCL [42] approach that uses supervised contrastive learning to assist classification. Additionally, in all six scenarios of the Deepfake dataset, we extend our comparison to include manual-feature-based methods encompassing FastText [4] and GLTR [22], in addition to DetectGPT [47], a statistical-based method.

**Implementation details.** For all our method's experiments, we use the interfaces and pre-trained model weights from the HuggingFace transformers [28] library. We freeze the embedding layers and only train the remaining model parameters. All experiments use the same hyperparameters and an AdamW [44] optimizer with a cosine annealing learning rate schedule. The peak learning rate is set at 2e-05, warmed up linearly for 2000 steps, and weight decay is set to 1e-04. The maximum input token length is 512. We train for 50 epochs with batch size of 32 per GPU on 8 NVIDIA V100 GPUs. During inference, we implement with an efficient K-Nearest Neighbors (KNN) [15] algorithm provided by the Faiss [46] library, to perform classification. For all comparative experiments, we use their open-source code and default settings for training and testing, and then report the results.

## 4.2 Main Results

Firstly, we fine-tune multiple pre-trained text encoders on Cross-domains & Cross-models subset of the Deepfake [39] dataset using our method to validate its broad compatibility. As shown in Table 6, all models improve on their baselines, confirming our method's effectiveness with diverse text encoders in AI-generated text detection. Among them, the SimCSE-RoBERTa [21] model achieves the second-best performance with relatively fewer parameters. Thus, we select this model as our text encoder for all the subsequent experiments.

Subsequently, to validate the performance in comparison to existing approaches, and to ascertain its robustness, we conduct experiments on three commonly-used datasets. These include the M4 [68] dataset (M4-monolingual and M4-multilingual), TuringBench [61], and the Cross-domains & Cross-models subset of Deepfake which is the largest and most challenging subset in the In-distribution scenarios of Deepfake. The results are shown in Table 1. Our method achieves the state-of-the-art performance on each dataset. Using the AvgRec metric for illustration, our method surpasses the second-best method by 6.52% in the M4-monolingual setting and by 7.15% in the M4-multilingual setting. Despite the comparatively lower difficulty of the earlier released TuringBench dataset, where all comparative methods perform well, our model still outperforms the second-best by 0.15%. Furthermore, in the Cross-domains & Cross-models subset of Deepfake, our method exceeds the runner-up by 2.66%. Indicated by the aforementioned experimental results, our method performs commendably across multiple datasets, demonstrating that the framework we propose is robust against diverse data distributions and scenarios.

To verify the capability of our method in terms of domain adaptation and out-of-distribution (OOD) detection, we conduct experiments on all six scenarios proposed in the Deepfake dataset. The dataset is strictly divided into different subsets to ensure that the testing data used for any given scenario is not used as training data for other settings. In In-distribution detection, comparison methods are trained

Table 1: Experimental results on M4-monolingual [68], M4-multilingual [68], TuringBench [61] and Deepfake's Cross-domains & Cross-models subset [39]. The best number is highlighted in **bold**, while the second best one is underlined.

| Method | M4-monolingual | | M4-multlingual | | TuringBench | | Deepfake | |
|---|---|---|---|---|---|---|---|---|
| | AvgRec | F1 | AvgRec | F1 | AvgRec | F1 | AvgRec | F1 |
| RoBERTa | 88.70 | 88.44 | 80.01 | 84.44 | 99.59 | 99.29 | 87.30 | 88.37 |
| SCL (ICLR 2021) | 91.92 | 91.21 | 86.27 | 84.75 | 99.46 | 99.22 | 90.59 | 89.83 |
| Longformer (ACL 2024) | 80.99 | 81.42 | 84.68 | 83.00 | 99.40 | 98.95 | 90.53 | 89.76 |
| T5-Sentinel (EMNLP 2023) | 84.01 | 81.08 | 76.21 | 68.99 | 99.39 | 97.43 | 93.49 | 93.30 |
| Binoculars (ICML 2024) | 89.89 | 89.89 | 80.63 | 82.43 | 51.24 | 9.98 | 64.96 | 70.58 |
| **DeTeCTive (Ours)** | **98.44** | **98.38** | **93.42** | **93.05** | **99.74** | **99.35** | **96.15** | **96.16** |

Table 2: Experimental results of AvgRec on six scenarios proposed in Deepfake [39] dataset. In Out-of-distribution detection, our method produces two results. The left one is the regular testing result while the right one indicates the result combining with TFIA. The best number is highlighted in **bold**, while the second best one is underlined. For detailed results, please refer to Table 12.

| Detection Scenario | Testbed Type | Longformer | GLTR | DetectGPT | FastText | DeTeCtive (Ours) |
|---|---|---|---|---|---|---|
| In-distribution | Cross-domains & Cross-models | 90.53 | 55.42 | 60.48 | 78.80 | **96.15** |
| | Cross-domains & Model-specific | 96.10 | 77.58 | 62.31 | 83.02 | **96.73** |
| | Domain-specific & Cross-models | 93.51 | 63.08 | 60.48 | 81.67 | **96.11** |
| | Domain-specific & Model-specific | 96.60 | 87.45 | 86.37 | 94.54 | **99.77** |
| Out-of-distribution | Unseen Models | 86.61 | 57.49 | 62.31 | 68.61 | 92.19/**93.03** |
| | Unseen Domains | 68.40 | 56.48 | 60.48 | 63.54 | 82.60/**89.63** |

separately on each specific subset and then averaged to get the final results. Conversely, we only train on the Cross-domains & Cross-models subset. During testing, we solely employ each scenario's training data as the database, skipping additional training on these data and progressing directly to inference. Our method outperforms other methods in every setting. The precise experimental results of AvgRec are presented in the first row of Table 2. For the Out-of-distribution detection, it is further divided into two cases: Unseen Models and Unseen Domains. The testing set includes data from the above two scenarios, which has not appeared in the training set. The AvgRec results are as shown in the second row of Table 2, where our method surpasses the next by 5.58% and 14.2% respectively in terms of AvgRec. The results demonstrate the good generalization performance of our method, considerably outperforming existing methods. Finally, we devise a set of experiments where we incorporate corresponding OOD data from training sets of the Cross-domains & Cross-models subset into the database to aid detection. There is a substantial performance improvement in the Unseen Domains scenario, with an additional 7.03% increase in AvgRec. For the Unseen Models, only a slight improvement is observed, which can be attributed to the existing capability of identifying similar models. This also highlights the effectiveness of the multi-level contrastive learning within our method from another perspective. We refer to this finding as Training-Free Incremental Adaptation (TFIA), and we delve deeper into the analysis of TFIA capability in Section 4.4.

### 4.3 More Applications

**Attack robustness.** In order to investigate the robustness of our method to paraphrasing attack, we conduct experiments on the OUTFOX [33] dataset. The experiments are divided into three scenarios: Non-attacked, DIPPER [35] attack, and OUTFOX attack, the results are presented in Table 3. From the experimental results, it can be seen that our method achieves the best results under all three settings, and the performance of our method does not decline much after being attacked, whereas the performance of other methods declines significantly. The analysis is as follows, we believe that our usage of the K-Nearest Neighbours (KNN) algorithm for classification offers our approach with a level of fault tolerance. Thus, minor disturbances prompted by certain attacks do not engender significant feature drift. Consequently, our method remains effective in detection. Therefore, these experiments show that our method has good robustness against paraphrasing attack.

**Authorship attribution detection.** To further probe the efficacy of our method in the task of authorship attribution detection, we conduct comprehensive experiments on TuringBench [35] dataset,

Table 3: Experimental results on attack robustness on OUTFOX [33] dataset, including DIPPER [35] attack and OUTFOX attack methods. The best number is highlighted in **bold**.

| Attacker | Non-attacked | | DIPPER | | OUTFOX | |
|---|---|---|---|---|---|---|
| Detector | AvgRec | F1 | AvgRec | F1 | AvgRec | F1 |
| RoBERTa-base | 93.0 | 92.9 | 91.5 | 91.3 | 81.5 | 78.9 |
| RoBERTa-large | 90.8 | 90.7 | 94.3 | 94.4 | 73.9 | 68.3 |
| HC3 Detector | 74.9 | 73.8 | 41.3 | 5.5 | 39.8 | 0.7 |
| OUTFOX | 96.5 | 96.4 | 82.4 | 79.0 | 61.8 | 39.4 |
| **DeTeCTive (Ours)** | **99.1** | **99.1** | **97.7** | **97.5** | **97.0** | **96.9** |

Table 4: Experimental results of authorship attribution detection on TuringBench [61] dataset. The best number is highlighted in **bold**.

| Method | Precision | Accuracy | Recall | F1 |
|---|---|---|---|---|
| Random Forest | 58.93 | 61.47 | 60.53 | 58.47 |
| SVM (3-grams) | 71.24 | 72.99 | 72.23 | 71.49 |
| WriteprintsRFC | 45.78 | 49.43 | 48.51 | 46.51 |
| Syntax-CNN | 65.20 | 66.13 | 65.44 | 64.80 |
| N-gram CNN | 69.09 | 69.14 | 68.32 | 66.65 |
| N-gram LSTM | 66.94 | 68.98 | 68.24 | 66.46 |
| OpenAI Detector | 78.10 | 78.73 | 78.12 | 77.41 |
| BertAA | 77.96 | 78.12 | 77.50 | 77.58 |
| BERT-Multinomial | 80.31 | 80.78 | 80.21 | 79.96 |
| RoBERTa-Multinomial | 82.14 | 81.73 | 81.26 | 81.07 |
| **DeTeCtive (Ours)** | **84.04** | **82.75** | **82.59** | **83.05** |

comparing our method against various baseline solutions. As depicted in Table 4, our method illustrates commendable performance in this task, substantiating its capacity to learn and apply multi-level features effectively in a multi-class classification context.

## 4.4 Ablation studies and Analysis

**Ablation studies.** To systematically evaluate the effects of each component in our method, we conduct a series of ablation studies as shown in Table 5. The experiments show that removing any loss term results in a performance decrease. Notably, when the multi-level contrastive loss $\mathcal{L}_{mcl}$ in Eq. 9 we proposed is replaced by a plain contrastive loss $\mathcal{L}_{pcl}$, the performance declines the most compared to other loss terms, because only the human-written text and AI-generated text are treated as negative sample pairs, without considering the internal relations. Furthermore, using a similarity-based KNN classification scheme also enhances the performance.

**Analysis on TFIA.** We further explore how incrementally adding corresponding OOD samples affects the performance, illustrated in Figure 2. The results demonstrate that as more OOD data are incorporated into the database, the model's performance improves consistently. Adding a modest amount of OOD data can considerably enhance the performance, particularly noticeable in unseen

Table 5: Ablation studies on loss design and classification approach, all conducted on Deepfake's Cross-domains & Cross-models subset [39].

| Ablation Components | Configurations | HumanRec | MachineRec | AvgRec* | F1* |
|---|---|---|---|---|---|
| | $\mathcal{L}_{all}$ (Baseline) | 95.36 | 96.94 | **96.15** | **96.16** |
| | $\mathcal{L}_{pcl} + \mathcal{L}_{ce}$ | 91.93 | 96.51 | 94.22 | 94.12 |
| Loss desgin | w/o $\mathcal{L}_{ce}$ | 93.03 | 96.99 | 95.01 | 94.95 |
| (classification w/ KNN) | w/ $\alpha = 0$ | 93.89 | 96.61 | 95.25 | 95.22 |
| | w/ $\beta = 0$ | 92.85 | 97.03 | 94.94 | 94.87 |
| | w/ $\gamma = 0$ | 92.89 | 96.86 | 94.88 | 94.81 |
| Classification approach | w/ classification head | 88.99 | 97.39 | 93.19 | 92.92 |

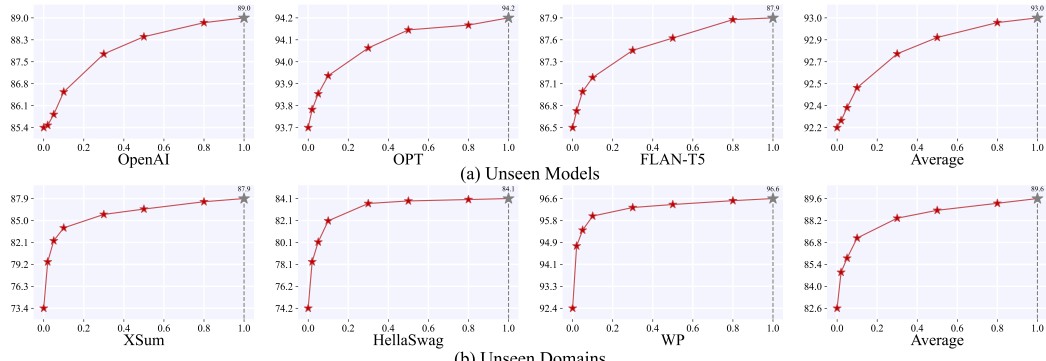

Figure 2: Analysis of model performance changes with the addition of OOD data. The x-axis represents the proportion of OOD data added, and the y-axis represents the AvgRec metric. (a) presents the results for Unseen Models, and (b) for Unseen Domains.

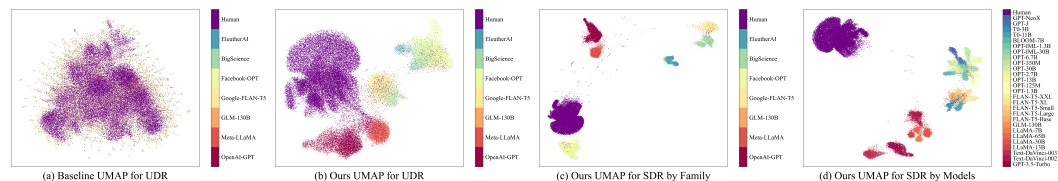

Figure 3: UMAP [45] dimensionality reduction visualization results, Where UDR stands for Unsupervised Dimensionality Reduction and SDR stands for Supervised Dimensionality Reduction.

domain scenarios. This suggests that in practical applications, TFIA could effectively mitigate the unsatisfactory adaptability of current methods to OOD data. For more detailed information about the TFIA experiments, please refer to Appendix E.

**Visualizations of learned embeddings.** To further verify our method's capability to differentiate various writing styles, we apply UMAP [45] for dimensionality reduction on text embeddings from the test set of the Deepfake Cross-domains & Cross-models subset. As shown in Figure 3 (a), using a pre-trained model directly fails to segregate embeddings of varying categories. In contrast, after fine-tuning with our method, UMAP unsupervised dimensionality reduction is already capable of clustering the features of various categories well, as shown in Figure 3 (b). With UMAP supervised dimensionality reduction, as shown in Figure 3 (c) and (d), our model further reflects the multi-level relations either between model families or individual models.

## 5 Conclusion

In this paper, we propose **DeTeCtive**, a novel method for AI-generated text detection, anchored by a multi-task auxiliary multi-level contrastive learning framework. Through extensive experiments, our method demonstrates state-of-the-art performance on three popular benchmarks, validating the effectiveness of each component via ablation studies. We also uncover our method's Training-Free Incremental Adaptation (TFIA) capability, enriching its experimental analysis. We hope our work brings new insights and findings for the task of AI-generated text detection.

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

# A   Limitation and Future Work

In this paper, we have not thoroughly explored our method's interpretability, which we believe is a promising research direction. Follow-up research works can analyze the differences and similarities between human-written text and AI-generated text based on our open-source model, and conduct token-level interpretability research. Also, we have not carried out training on a larger corpus. We believe that performing our method on a larger corpus could enhance the ability to identify writing styles, thereby improving the model's performance.

# B   Broader Impacts

The topic of this study is AI-generated text detection, a subject of significant importance for AI-safety. With the rapid development of AI technology, particularly in Natural Language Processing (NLP), the proliferation of AI-generated text raises concerns about global information security, as it may contribute to the spread of disinformation, false information, and content that has the potential to encourage harmful or destructive behaviors, making the detection and monitoring of AI-generated text a pressing issue. The method presented in this paper achieves state-of-the-art performance on several benchmarks. Particularly noteworthy is its superior performance in out-of-distribution (OOD) detection, far surpassing existing methods. These advancements offer promising prospects for the real-world application of AI-generated text detection algorithms. The methodological advancements in this research endeavor to facilitate the safe and ethical usage of AI technologies, consequently strengthening societal security.

# C   Dataset Details

## C.1   Deepfake

The Deepfake [39] dataset collects human-written text from 10 domains. The AI-generated texts are produced by 27 LLMs and have been categorized into 7 model sets, as shown in Table 7. These texts are generated by three types of prompts: continuation prompts, topical prompts, and specified prompts. Table 8 details the specific sources of the dataset and the splits of training, validation, and testing sets. The Deepfake dataset contains 6 different scenarios, carefully divided to ensure that the testing data used for any specific scenario would not be used as training data for other scenarios. These scenarios are categorized into: In-distribution detection and Out-of-distribution detection as follows.

**In-distribution detection.**   In-distribution detection scenario includes four subsets:

- **Domain-specific & Model-specific.** Human-written texts come from a specific domain, and AI-generated texts are from a specific GPT-J-6B [64] model. There are 10 testbeds based on different domains.

- **Domain-specific & Cross-models.** Human-written texts come from a specific domain, and AI-generated texts are from different models. There are 10 testbeds based on different domains.

- **Cross-domains & Model-specific.** Human-written texts are from different domains, and AI-generated texts are from a specific model set. There are 7 testbeds based on different model sets.

- **Cross-domains & Cross-models.** All models and domains are mixed to create a general subset.

**Out-of-distribution detection.**   Out-of-distribution detection scenario includes two subsets:

- **Unseen models.** Texts generated by a specific model set are excluded from the training set. Testing data only comes from the excluded model set. There are 7 testbeds based on different excluded model sets.

- **Unseen domains.** Texts from a specific domain are excluded from the training set, with training data containing text from other domains. Testing data only comes from the excluded domain. There are 10 testbeds based on different excluded domains.

## C.2 M4

The M4 [68] dataset is a large-scale dataset featuring multi-domain, multi-model, and multilingual characteristics, as shown in Table 9 and Table 10. This dataset includes text from Wikipedia, WikiHow [34], Reddit [19], arXiv, and PeerRead [29]. Using human-written prompts, models like ChatGPT [6], DaVinci-003 [6], LLaMA [60], FLAN-T5 [13], Cohere [36], Dolly-v2 [14], and BLOOMz [49] generate text in 9 different languages, including English, Chinese, and Russian. [67] organizes a competition to detect AI-generated text based on M4, including tasks such as whole-paragraph detection and sentence-level detection. We use two scenarios designed for whole-paragraph detection: monolingual and multilingual. In monolingual scenario, the test set includes unseen AI-generated texts from GPT-4 [1], and are paraphrased by OUTFOX [33], which increases the difficulty for detection. In multilingual scenario, the test set contains novel languages that have not appeared in either training set or validation set, and AI-generated texts are also paraphrased.

## C.3 TuringBench

The TuringBench [61] dataset provides a benchmark for systematically evaluating AI-generated text detection. The human-written texts come from news titles and contents from CNN, Washington Post, and Kaggle. Using these article titles, various LLMs, including the GPT series [6], GROVER series [53], CTRL [31], XLM [37], and XLNET [72], generate articles similar to human-written text, resulting in 200K articles with 20 labels, detailed in Table 11.

# D   Experiments on compatibility with diverse encoders

The results of fine-tuning various text encoders using our approach are shown in Table 6.

Table 6: Experimental results of applying our method to multiple text encoders on Cross-domains & Cross-models subset of the Deepfake [39] dataset. The best number is highlighted in **bold**, while the second best one is underlined.

| Text Encoders | Params | Baseline Results | | Fine-tuned Results | |
|---|---|---|---|---|---|
| | | AvgRec | F1 | AvgRec | F1 |
| E5$_{base}$ [65] | 109M | 72.88 | 73.57 | 94.65 (+21.77) | 94.73 (+21.16) |
| BGE$_{base}$ [71] | 109M | 64.41 | 61.66 | 93.95 (+29.54) | 93.89 (+32.23) |
| GTE$_{large}$ [40] | 335M | 65.77 | 62.55 | 95.20 (+29.43) | 95.23 (+32.68) |
| BERT$_{base}$ [18] | 109M | 77.43 | 76.61 | 94.53 (+17.10) | 94.46 (+17.85) |
| BERT$_{large}$ | 335M | 75.65 | 74.85 | 95.32 (+19.67) | 95.37 (+20.52) |
| RoBERTa$_{base}$ [43] | 125M | 77.91 | 76.93 | 95.04 (+17.13) | 94.98 (+18.05) |
| FLAN-T5$_{base}$ [13] | 223M | 66.37 | 65.53 | 95.52 (+29.15) | 95.48 (+29.95) |
| FLAN-T5$_{large}$ | 750M | 67.46 | 66.72 | **96.53** (+29.07) | **96.53** (+29.81) |
| AnglE-BERT$_{large}$ [38] | 335M | 63.59 | 61.14 | 94.66 (+31.44) | 94.70 (+33.56) |
| SimCSE-BERT$_{base}$ | 109M | 68.22 | 68.96 | 94.13 (+25.91) | 94.00 (+25.04) |
| SimCSE-RoBERTa$_{base}$ [21] | 125M | 66.44 | 64.36 | 96.15 (+29.71) | 96.16 (+31.80) |

# E   Detailed Description of Experiments on Deepfake dataset

Here, we provide a detailed description of experiments conducted under the six scenarios proposed in Deepfake [39] dataset.

In In-distribution detection, Longformer [2] is trained separately on all testbeds of each specific subset, with the final results averaged. FastText [4], GLTR [22], and DetectGPT [47] are directly tested on all testbeds of each subset based on statistical features. Our model is solely trained on the Cross-domains & Cross-models subset, while for the other three subsets, we use only the testbed training data from each subset as the database for inference without additional training. Notably, the Deepfake dataset is strictly divided to ensure that data used for a specific testing scenario is not used as training data in other scenarios.

In Out-of-distribution (OOD) detection, Longformer is trained separately on all testbeds of each specific subset, with the final results averaged. FastText, GLTR, and DetectGPT are directly tested on all testbeds of each test scenario based on statistical features, also with the final results averaged. We train on the training set provided for each testbed, and use these data as the database for testing. The final results are obtained by averaging all the scores. Our method's detailed results for different testbeds in each scenario are shown in Table 13.

To validate the model's Training-Free Incremental Adaptation (TFIA) capability, we design the following experiments. Firstly, it is worth noting that the Deepfake dataset includes data from 10 domains and 7 model sets. In each OOD testbed, the training set excludes data from a specific domain or model set, while the test set consists of data from the domain or model set that is excluded in the training set. For example, in unseen models, there are 7 testbeds. For the first testbed of unseen models in Table 13, the training set excludes data from the LLaMA series, while the test set is composed of data from the LLaMA model set. In OOD testing, the training data from the remaining six model sets is used as the database for testing. To validate the TFIA capability, we add training data from the LLaMA model set to the OOD database, noting that the added training data does not appear in the test set, ensuring that no testing data leakage occurred during our testing.

Additionally, in Figure 2, we further explore the TFIA capability by gradually adding unseen model or domain data into the database, analyzing the impact of the added data quantity on model performance. For the Unseen-Domains-XSum testbed of unseen domains in Table 13, we gradually add XSum-domain training data at increments of 5%, 10%, and 15% until the training set data for that domain is exhausted, reaching a ratio of 100%.

Table 7: Models included in Deepfake [39].

| Model Set | Models |
|---|---|
| OpenAI GPT [6] | GPT-3.5-Turbo, Text-DaVinci-002, Text-DaVinci-003 |
| Meta LLaMA [60] | LLaMA-13B, LLaMA-30B, LLaMA-65B, LLaMA-7B |
| Facebook OPT [76] | OPT-125M,OPT-350M, OPT-1.3B, OPT-IML-Max-1.3B, OPT-2.7B
OPT-6.7B, OPT-13B, OPT-30B, OPT-IML-30B |
| GLM-130B [74] | GLM-130B |
| Google FLAN-T5 [13] | FLAN-T5-Small,FLAN-T5-Base, FLAN-T5-Large
FLAN-T5-XL,FLAN-T5-XXL |
| BigScience | BLOOM-7B [49], T0-3B [54], T0-11B |
| EleutherAI | GPT-J [64], GPT-NeoX [3] |

Table 8: The specific origins and splits of Deepfake [39].

| Dataset | CMV [57] | Yelp [77] | XSum [50] | TLDR | ELI5 [19] |
|---|---|---|---|---|---|
| Train | 4,461/21,130 | 32,321/21,048 | 4,729/26,372 | 2,832/20,490 | 17,529/26,272 |
| Valid | 2,549/2,616 | 2,700/2,630 | 3,298/3,297 | 2,540/2,520 | 3,300/3,283 |
| Test | 2,431/2,531 | 2,685/2,557 | 3,288/3,261 | 2,536/2,451 | 3,193/3,215 |
| WP [20] | ROC [48] | HellaSwag [73] | SQuAD [52] | SciGen [8] | all |
| 6,768/26,339 | 3,287/26,289 | 3,129/25,584 | 15,905/21,489 | 4,644/21,541 | 95,596/236,554 |
| 3,296/3,288 | 3,286/3,288 | 3,291/3,190 | 2,536/2,690 | 2,671/2,670 | 29,467/29,462 |
| 3,243/3,192 | 3,275/3,207 | 3,292/3,078 | 2,509/2,535 | 2,563/2,338 | 29,015/28,365 |

Table 9: Data statistics of M4 **Monolingual** setting over Train/Dev/Test splits.

| Split | Source | DaVinci-003 | ChatGPT | Cohere | Dolly-v2 | BLOOMz | GPT-4 | Machine | Human |
|---|---|---|---|---|---|---|---|---|---|
| Train | Wikipedia | 3,000 | 2,995 | 2,336 | 2,702 | - | - | 11,033 | 14,497 |
| | Wikihow | 3,000 | 3,000 | 3,000 | 3,000 | - | - | 12,000 | 15,499 |
| | Reddit | 3,000 | 3,000 | 3,000 | 3,000 | - | - | 12,000 | 15,500 |
| | arXiv | 2,999 | 3,000 | 3,000 | 3,000 | - | - | 11,999 | 15,498 |
| | PeerRead | 2,344 | 2,344 | 2,342 | 2,344 | - | - | 9,374 | 2,357 |
| Dev | Wikipedia | - | - | - | - | 500 | - | 500 | 500 |
| | Wikihow | - | - | - | - | 500 | - | 500 | 500 |
| | Reddit | - | - | - | - | 500 | - | 500 | 500 |
| | arXiv | - | - | - | - | 500 | - | 500 | 500 |
| | PeerRead | - | - | - | - | 500 | - | 500 | 500 |
| Test | Outfox | 3,000 | 3,000 | 3,000 | 3,000 | 3,000 | 3,000 | 18,000 | 16,272 |

Table 10: Data statistics of M4 **Multilingual** setting over Train/Dev/Test splits.

| Split | Language | DaVinci-003 | ChatGPT | LLaMA 2 | Jais | Other | Machine | Human |
|-------|----------|-------------|---------|---------|------|-------|---------|-------|
| Train | English | 11,999 | 11,995 | - | - | 35,036 | 59,030 | 62,994 |
| | Chinese | 2,964 | 2,970 | - | - | - | 5,934 | 6,000 |
| | Urdu | - | 2,899 | - | - | - | 2,899 | 3,000 |
| | Bulgarian | 3,000 | 3,000 | - | - | - | 6,000 | 6,000 |
| | Indonesian | - | 3,000 | - | - | - | 3,000 | 3,000 |
| Dev | Russian | 500 | 500 | - | - | - | 1,000 | 1,000 |
| | Arabic | - | 500 | - | - | - | 500 | 500 |
| | German | - | 500 | - | - | - | 500 | 500 |
| Test | English | 3,000 | 3,000 | - | - | 9,000 | 15,000 | 13,200 |
| | Arabic | - | 1,000 | - | 100 | - | 1,100 | 1,000 |
| | German | - | 3,000 | - | - | - | 3,000 | 3,000 |
| | Italian | - | - | 3,000 | - | - | 3,000 | 3,000 |

Table 11: The number of data samples generated by each generator in TuringBench [61].

| Text Generator | Data samples |
|----------------|--------------|
| Human | 8,854 |
| GPT-1 | 8,309 |
| GPT-2_small | 8,164 |
| GPT-2_medium | 8,164 |
| GPT-2_large | 8,164 |
| GPT-2_xl | 8,309 |
| GPT-2_PyTorch | 8,854 |
| GPT-3 | 8,164 |
| GROVER_base | 8,854 |
| GROVER_large | 8,164 |
| GROVER_mega | 8,164 |
| CTRL | 8,121 |
| XLM | 8,852 |
| XLNET_base | 8,854 |
| XLNET_large | 8,134 |
| FAIR_wmt19 | 8,164 |
| FAIR_wmt20 | 8,309 |
| TRANSFORMER_XL | 8,306 |
| PPLM_distil | 8,854 |
| PPLM_gpt2 | 8,854 |

Table 12: The detailed results on six scenarios of Deepfake [39] dataset. The best number is highlighted in **bold**, while the second best one is underlined. In the table, the value of N/A indicates that we are unable to infer specific results based on the data from the Deepfake paper [39]. The notation "w/C&C database" represents the results combined with TFIA.

| Settings | Methods | HumanRec | MachineRec | AvgRec$^*$ | F1$^*$ |
|---|---|---|---|---|---|
| **In-distribution Detection** | | | | | |
| **Domain-specific & Model-specific** | FastText | 94.72 | 94.36 | 94.54 | N/A |
| | GLTR | 90.96 | 83.94 | 87.45 | N/A |
| | Longformer | 97.30 | 95.91 | 96.60 | N/A |
| | DetectGPT | 91.68 | 81.06 | 86.37 | N/A |
| | **DeTeCtive (ours)** | 99.78 | 99.77 | **99.77** | **99.79** |
| **Cross-domains & Model-specific** | FastText | 88.96 | 77.08 | 83.02 | N/A |
| | GLTR | 75.61 | 79.56 | 77.58 | N/A |
| | Longformer | 95.25 | 96.94 | 96.10 | N/A |
| | DetectGPT | 48.67 | 75.95 | 62.31 | N/A |
| | **DeTeCtive (ours)** | 96.51 | 96.95 | **96.73** | **96.73** |
| **Domain-specific & Cross-models** | FastText | 89.43 | 73.91 | 81.67 | N/A |
| | GLTR | 37.25 | 88.90 | 63.08 | N/A |
| | Longformer | 89.78 | 97.24 | 93.51 | N/A |
| | DetectGPT | 86.92 | 34.05 | 60.48 | N/A |
| | **DeTeCtive (ours)** | 95.16 | 97.06 | **96.11** | **96.11** |
| **Cross-domains & Cross-models** | FastText | 86.34 | 71.26 | 78.80 | 80.53 |
| | GLTR | 12.42 | 98.42 | 55.42 | 21.80 |
| | Longformer | 82.80 | 98.27 | 90.53 | 89.76 |
| | DetectGPT | 86.92 | 34.05 | 60.48 | 69.16 |
| | **DeTeCtive (ours)** | 95.36 | 96.94 | **96.15** | **96.16** |
| **Out-of-distribution Detection** | | | | | |
| **Unseen Models** | FastText | 83.12 | 54.09 | 68.61 | N/A |
| | GLTR | 25.77 | 89.21 | 57.49 | N/A |
| | Longformer | 83.31 | 89.09 | 86.61 | N/A |
| | DetectGPT | 48.67 | 75.95 | 62.31 | N/A |
| | **DeTeCtive (ours)** | 93.90 | 90.48 | 92.19 | 92.46 |
| | **w/ C&C database** | 92.69 | 93.36 | **93.03** | **93.05** |
| **Unseen Domains** | FastText | 54.29 | 72.79 | 63.54 | N/A |
| | GLTR | 15.84 | 97.12 | 56.48 | N/A |
| | Longformer | 38.05 | 98.75 | 68.40 | N/A |
| | DetectGPT | 86.92 | 34.05 | 60.48 | N/A |
| | **DeTeCtive (ours)** | 68.22 | 96.99 | 82.60 | 76.73 |
| | **w/ C&C database** | 84.09 | 95.17 | **89.63** | **88.74** |

Table 13: Detailed results on all testbeds in each scenario of Deepfake [39] dataset.

| Settings | Sub-settings | HumanRec | MachineRec | AvgRec* | F1* |
|---|---|---|---|---|---|
| **In-distribution Detection** | | | | | |
| **Domain-specific & Model-specific** | CMV | 100.0 | 100.0 | 100.0 | 100.0 |
| | ELI5 | 100.0 | 98.89 | 99.44 | 99.51 |
| | HellaSwag | 99.05 | 100.0 | 99.52 | 99.52 |
| | ROC | 100.0 | 100.0 | 100.0 | 100.0 |
| | Scigen | 100.0 | 100.0 | 100.0 | 100.0 |
| | SQuAD | 100.0 | 100.0 | 100.0 | 100.0 |
| | TLDR | 98.73 | 100.0 | 99.37 | 99.36 |
| | WP | 100.0 | 100.0 | 100.0 | 100.0 |
| | XSum | 100.0 | 100.0 | 100.0 | 100.0 |
| | Yelp | 100.0 | 98.78 | 99.39 | 99.51 |
| | **Average** | **99.78** | **99.77** | **99.77** | **99.79** |
| **Cross-domains & Model-specific** | LLaMA | 95.42 | 96.87 | 96.15 | 96.12 |
| | BigScience | 97.07 | 97.77 | 97.42 | 97.42 |
| | FLAN-T5 | 96.39 | 93.07 | 94.73 | 94.82 |
| | GLM-130B | 94.46 | 95.86 | 95.16 | 95.13 |
| | EleutherAI | 98.62 | 99.65 | 99.14 | 99.13 |
| | OpenAI | 95.59 | 97.00 | 96.29 | 96.27 |
| | OPT | 97.99 | 98.44 | 98.22 | 98.21 |
| | **Average** | **96.51** | **96.95** | **96.73** | **96.73** |
| **Domain-specific & Cross-models** | CMV | 96.92 | 98.77 | 97.85 | 97.80 |
| | ELI5 | 94.52 | 95.18 | 94.85 | 94.81 |
| | HellaSwag | 93.48 | 97.71 | 95.59 | 95.58 |
| | ROC | 94.99 | 96.48 | 95.74 | 95.75 |
| | Scigen | 95.28 | 98.71 | 96.99 | 97.01 |
| | SQuAD | 96.58 | 96.88 | 96.73 | 96.73 |
| | TLDR | 90.16 | 97.75 | 93.96 | 93.76 |
| | WP | 98.55 | 99.55 | 99.05 | 99.04 |
| | XSum | 94.28 | 98.86 | 96.57 | 96.50 |
| | Yelp | 96.79 | 90.73 | 93.76 | 94.11 |
| | **Average** | **95.16** | **97.06** | **96.11** | **96.11** |
| **Cross-domains & Cross-models** | **Average** | **95.36** | **96.94** | **96.15** | **96.16** |
| **Out-of-distribution Detection** | | | | | |
| **Unseen models** | LLaMA | 94.45 | 93.93 | 94.19 | 94.21 |
| | BigScience | 93.81 | 94.76 | 94.28 | 94.26 |
| | FLAN-T5 | 93.24 | 79.85 | 86.54 | 87.38 |
| | GLM-130B | 94.35 | 94.56 | 94.45 | 94.45 |
| | EleutherAI | 93.91 | 99.72 | 96.82 | 96.72 |
| | OpenAI | 94.50 | 76.26 | 85.38 | 86.60 |
| | OPT | 93.04 | 94.30 | 93.67 | 93.63 |
| | **Average** | **93.90** | **90.48** | **92.19** | **92.46** |
| **Unseen domains** | CMV | 93.55 | 97.37 | 95.46 | 95.32 |
| | ELI5 | 81.12 | 96.87 | 88.99 | 88.04 |
| | HellaSwag | 54.59 | 93.81 | 74.20 | 68.09 |
| | ROC | 23.15 | 99.03 | 61.09 | 37.30 |
| | Scigen | 84.49 | 95.69 | 90.09 | 89.73 |
| | SQuAD | 68.26 | 98.48 | 83.37 | 80.41 |
| | TLDR | 69.03 | 95.74 | 82.39 | 69.03 |
| | WP | 87.35 | 97.55 | 92.45 | 92.02 |
| | XSum | 49.92 | 96.80 | 73.36 | 65.22 |
| | Yelp | 70.71 | 98.51 | 84.61 | 82.15 |
| | **Average** | **68.22** | **96.99** | **82.60** | **76.73** |

