# OpenReview forum: "DeTeCtive: Detecting AI-generated Text via Multi-Level Contrastive Learning"
_NeurIPS.cc/2024/Conference — NeurIPS 2024 poster_

### Official Review · Reviewer_Tv1c · 2024-06-22

**Soundness:** 2
**Presentation:** 3
**Contribution:** 2
**Rating:** 5
**Confidence:** 4

**Summary:**

This paper presents DeTeCtive, a new algorithm for detecting AI-generated text. The key insight in this paper is that instead of treating AI-generated text detection as a binary classification problem, it should be treated as a multi-class "author style" classification problem. Using this insight, the authors develop a contrastive learning algorithm which learns style vectors for each LLM style, which are then stored in a style vector database for inference via KNN classification. The learnt style vector database can easily be expanded at inference to accomodate new text generated from potentially OOD LLMs.

The authors perform experiments on 3 existing datasets with outputs from 8 to 27 different LLMs, and show that DeTeCtive outperforms some existing methods. The authors complement their work with ablation studies on their loss function, and analysis experiments in OOD settings.

**Strengths:**

1. The paper presents an interesting reformulation of AI-generated text detection, as a multi-label "LLM style" classification problem. The paper presents an intuitive contrastive learning algorithm to learn style vectors for each LLM style, which are then stored in a style vector database for inference.

2. It seems to be easy to adapt the proposed algorithm towards OOD LM-generated text. By simply computing vector representation on an OOD LLM's generated text, the algorithm can support AI-generated text detection on this OOD LLM in future cycles. The authors perform analysis experiments to confirm its effectiveness.

3. While I do have numerous concerns with the experiments, the authors do make an attempt at a large-scale empirical evaluation of their method, considering 3 existing datasets with outputs from 8 to 27 different LLMs.

**Weaknesses:**

I had some concerns about the experiments in this paper.

1. **Baselines seem to be quite weak / over a year old**. Most of the baselines used are either pre-2022 methods or DetectGPT, which seems unsuitable for a cross-model dataset like the one used in evaluation. How do newer methods like Binoculars [5] perform on the same benchmark? Also, why not compare against watermarking algorithms like KGW [1], EXP-Edit [2], SemStamp [3], or [4]? Also how well do commercial tools like GPTZero perform on the same datasets (https://gptzero.me)?

2. **Limited experiments on newer GPT-4 class LLMs**. AI-generated text detection is the hardest on the most human-like LLMs, which are the GPT-4 class models. However, almost no experiments were done on GPT-4 class models (except for a small subset of the M4 dataset). I encourage the authors to stress test their OOD generalization to outputs from GPT-4 class models like Claude Opus, Gemini 1.5 and the newer variants of GPT-4.

3. **Limited emphasis on attack robustness**. There are almost no experiments in the paper (besides the mention of OUTFOX paraphrases) showing the robustness of the DeTeCtive algorithm to paraphrasing [6, 7], text mixing attacks [8], and translation attacks. This is critical to establish the robustness of the method compared to alternative detectors.

4. **It's not clear whether gains over baselines are coming from the SimCSE initialization or training**. Looking at the ablation studies in Table 3 and baselines in Table 1, all ablated variants outperform all baselines. This makes me question the need for the complexity in the DeTeCtive method's loss function, and how much of the gains can be attributed to the SimCSE initialization rather than the proposed DeTeCtive algorithm. An ablation study that would help here is using the DeTeCtive loss function on alternative encoders like RoBERTa / BERT / SentenceBERT.

[1] - https://arxiv.org/abs/2301.10226
[2] - https://arxiv.org/abs/2307.15593
[3] - https://arxiv.org/abs/2310.03991
[4] - https://arxiv.org/abs/2305.08883
[5] - https://arxiv.org/pdf/2401.12070
[6] - https://arxiv.org/abs/2303.11156
[7] - https://arxiv.org/abs/2303.13408
[8] - https://openreview.net/pdf?id=DEJIDCmWOz

----

**After rebuttal**: I've decided to increase my score from 3 to 5 due to extra experiments on paraphrase robustness, and baseline comparisons against Binoculars.

**Questions:**

1. AI-generated text detectors typically need to operate in a low FPR setting, to minimize the risks of labeling innocent text as AI-generated. Given this, what are the Table 1 true positive rates at a low FPR of say 0-1%? TPR at low FPR ranges (0-1%) is a standard metric for evaluating AI-generated text detectors which has been used in many previous papers and blogs: https://arxiv.org/abs/2303.13408, https://openreview.net/pdf?id=DEJIDCmWOz, https://arxiv.org/pdf/2401.12070, https://foundation.mozilla.org/en/blog/who-wrote-that-evaluating-tools-to-detect-ai-generated-text/

**Limitations:**

The authors acknowledged weakness #3 in their limitations section. But I think this is a critical limitation and analysis on paraphrase / attack robustness is necessary in AI-generated text detection papers.

---

> ### Author Rebuttal · Authors · 2024-08-07
>
> *We would like to thank Reviewer Tv1c for the valuable feedback. The comments and suggestions have greatly helped us in improving the quality of our work. Please see below for our responses to your comments.*
> ***
> ## Baselines seem to be quite weak/over a year old.
> 1. We need to clarify that the comparison methods presented in our paper are commonly used baselines in the task of AI-generated text detection, and most of them are the latest state-of-the-art (SOTA) approaches **as the table shows in the top-level Author Rebuttal**. All the baseline schemes we identified are compared with DetectGPT. To align with related works in the field, we also include corresponding results.
> 2. Watermarking algorithms rely on embedding specific markers **during the generation process**. In contrast, our neural network based approach does not require any intervention in the generation process and can universally detect all texts. Since our goal is to provide a general detection method **independent of the generation process**, rather than intervening and controlling the model's generation process, our approach isn't suitable and feasible for comparison with watermarking algorithms. Moreover, watermarking algorithms are difficult to implement in closed-source models, and the flexible modifiability of open-source models also limits their widespread application.
> 3. The comparison with Binoculars and GPTZero can be respectively found at **the Table 2 and Table 3 within the uploaded PDF file**. The experimental results confirm that our method continues to outperform these two supplementary approaches.
> ***
> ## Limited experiments on newer GPT-4 class LLMs.
> In the out-of-distribution (OOD) detection experiment on the M4 dataset, a subset of the testing data is from GPT-4, and is utilized to evaluate the model's OOD detection capability. Our method demonstrates a performance advantage over comparative approaches on this dataset. Additionally, we supplement OOD experiments based on the "Unseen Domains & Unseen Models" dataset from the paper "MAGE: Machine-generated Text Detection in the Wild" presented at ACL 2024. **The testing data is sourced entirely from GPT-4**, and the results indicate that our method performs better than the comparative schemes. (Longformer† indicates that Longformer uses data from the testing set to determine the detection threshold.)
> | Methods | HumanRec | MachineRec | AvgRec | F1 |
> |---|---|---|---|---|
> | Unseen Domains & Unseen Model |  |  |  |  |
> | FastText | 71.78 | 68.88 | 70.33 | 70.21 |
> | GLTR | 16.79 | 98.63 | 57.71 | 28.40 |
> | Longformer | 52.50 | 99.14 | 75.82 | 68.45 |
> | Longformer† | 88.78† | 84.12† | 86.54† | 86.42 |
> | **DeTeCtive** | **75.46** | **97.88** | **86.67** | **84.93** |
> ***
> ## Limited emphasis on attack robustness.
> As you mentioned, a portion of the M4 dataset has undergone paraphrasing enhancement by OUTFOX, and our method outperforms all comparative solutions on this dataset. Additionally, we have supplemented our experiments with assessments of attack robustness, with the results shown **in the top-level Author Rebuttal and Table 4 within the uploaded PDF file**. The results demonstrate that our approach possesses strong resistance to attacks, indicating robust performance.
> ***
> ## It's not clear whether gains over baselines are coming from the SimCSE initialization or training.
> As described in **Section 4.2 of the paper**, we conduct experiments with **a variety of text encoders** to verify that our approach can enhance the performance of AI-generated text detection. **The results are already presented in Table 4 within the original submission**. This indicates that our method is not limited to a single encoder like SimCSE but is applicable to various encoders, and the performance gains come from our training.
> ***
> ## AI-generated text detectors typically need to operate in a low FPR setting, to minimize the risks of labeling innocent text as AI-generated.
> Our classification approach is based on dense retrieval and KNN (K-Nearest Neighbors) classification. This approach differs from traditional probability-based classifiers in that **it does not rely on probability outputs**. Specifically, the KNN classifier makes decisions based on the proximity of neighboring samples, rather than estimating the probability distribution of categories. Due to the absence of probability outputs, our method cannot define and adjust the threshold for the False Positive Rate (FPR). Typically, a low FPR range (such as 0-1%) is calculated by selecting different thresholds on the model's output probabilities, thereby measuring the True Positive Rate (TPR) at a given FPR. In our setup, the classification outcome is derived from the distance measurement of the nearest neighbors, **without any involvement of probability scoring or threshold adjustment**, providing only a global FPR and TPR. Therefore, we are unable to provide the corresponding TPR data within a specific low FPR range, such as 0-1%.

---

> > ### Comment · Reviewer_Tv1c · 2024-08-10
> > **Thank you for the detailed response and experiments, raising score to 5**
> >
> > Thank you for the detailed response and additional experiments! I've decided to increase my score from 3 to 5 due to extra experiments on paraphrase robustness, and baseline comparisons against Binoculars/GPTZero. I still encourage the authors to do add watermarking baselines in the paper, perhaps in the supplementary material.

---

> ### Author Response · Authors · 2024-08-12
>
> Thank you for your response and acknowledgment of our supplementary experiments. We are currently conducting experiments on watermarking dataset, this experiment requires a certain amount of time because we need to conduct evaluations on watermarking specific dataset, which entails some preliminary data processing work. We will include the results in the final version of the supplementary material within our paper.

---

### Official Review · Reviewer_W7ZQ · 2024-06-24

**Soundness:** 2
**Presentation:** 2
**Contribution:** 2
**Rating:** 5
**Confidence:** 4

**Summary:**

This paper proposes to learn a text encoder with contrastive learning to cluster texts from different sources for fine-grained classification. A training-free incremental adaptation method is designed for detecting OOD data.

**Strengths:**

1: The innovation of TFIA is inspiring for OOD detection. Detection in information retrieval style is interesting and exhibits great performance.

2: The solid experiments across multiple datasets prove the effectiveness of the proposed method.

**Weaknesses:**

1: The proposed multi-level contrastive loss shares some similarities with existing work [1] so it is not exciting enough.

2: Although the authors conduct extensive experiments on multiple datasets, only few of them are designed specifically for AI text detection (GLTR, DetectGPT). They should provide more comparisons with SOTA methods [2,3].

3:  As a work towards detecting AI-generated texts, more analysis of the features of texts should be provided. Otherwise, the framework could be applied to any binary classification problem (with hierarchical labels).

[1] Liu S, Liu X, Wang Y, et al. Does\textsc {DetectGPT} Fully Utilize Perturbation? Selective Perturbation on Model-Based Contrastive Learning Detector would be Better[J]. arXiv e-prints, 2024: arXiv: 2402.00263.

[2] Verma V, Fleisig E, Tomlin N, et al. Ghostbuster: Detecting text ghostwritten by large language models[J]. arXiv preprint arXiv:2305.15047, 2023.

[3] Chen Y, Kang H, Zhai V, et al. Gpt-sentinel: Distinguishing human and chatgpt generated content[J]. arXiv preprint arXiv:2305.07969, 2023.

**Questions:**

1: I am wondering if this framework would do a good job in authorship attribution [1]?

2: For the classification approach in ablation studies, is the classification head trained with your own data or initialized in other ways?

3: Could you please provide some explanation on the advantage of multi-level contrastive loss over pcl? It is confusing to me that fine-grained clustering could help with binary classification.

[1] Uchendu A, Le T, Shu K, et al. Authorship attribution for neural text generation[C]//Proceedings of the 2020 Conference on Empirical Methods in Natural Language Processing (EMNLP). 2020: 8384-8395.

**Limitations:**

The authors mention they do not explore the robustness of the proposed method. The validation could be conducted under different attacks like paraphrasing, editing, prompting.

---

> ### Author Rebuttal · Authors · 2024-08-07
>
> *We would like to thank Reviewer W7ZQ for the valuable feedback. The comments and suggestions have greatly helped us in improving the quality of our work. Please see below for our responses to your comments.*
> ***
> ## The proposed multi-level contrastive loss shares some similarities with existing work so it is not exciting enough.
> We would like to elaborate on the differences between our method and previous methods from the following aspects:
> 1. This paper utilizes a form of contrastive learning that adheres to the traditional definition of positive and negative sample pairs, which is also the plain contrastive learning(PCL) approach that we compared against in our paper. Their use of contrastive learning in conjunction with selective masking strategy aims to achieve better text representations. However, one of the core contributions of our work is the proposal of a multi-level contrastive learning(MCL) framework. We define multi-level positive and negative sample pairs and derive a loss function for multi-level contrastive learning, which is different from previous studies.
> 2. Additionally, we perform classification based on feature clustering, distinguishing it from traditional binary classification methods. Such a classification approach effectively enhances the model's detection capabilities and robustness.
> 3. Lastly, within the multi-level contrastive learning framework, our model also has the ability of Training-Free Incremental Adaptation, effectively improving out-of-distribution (OOD) detection capability. This is also an important capability that has not been explored or discovered in previous methods.
>
> Comprehensive ablation studies on our multi-level contrastive learning framework are performed in Section 4.3. From the experimental results, it can be seen that our method surpasses plain contrastive learning framework. Each component improves the model's performance, thereby validating the effectiveness of our proposed framework.
> ***
> ## Although the authors conduct extensive experiments on multiple datasets, only few of them are designed specifically for AI text detection (GLTR, DetectGPT). They should provide more comparisons with SOTA methods.
> Thanks for your affirmation of the amount of our experiments. We want to clarify that methods we compared across each dataset are commonly used baseline methods in the field of AI-generated text detection. The two comparison methods you referenced, Ghostbuster and GPT-Sentinel, have been factored into our research. In our paper, we make a comparison against T5-Sentinel, a latest work proposed by the author of GPT-Sentinel, demonstrating superior performance relative to GPT-Sentinel. Detailed results can be observed in **Table 1 of our paper**. As for Ghostbuster, due to the necessary utilization of OpenAI API interface for testing and because our test dataset is quite voluminous, it is not an appropriate choice for comparison. Instead, we select Binoculars as a stand-in comparison method,  which outperforms Ghostbuster in their paper. Our approach outperforms all compared methods; see **Table2 in the uploaded PDF** for details. Recently, an AI-generated text detection competition was held based on the M4 dataset. We also incorporate all the solutions from this competition into our list of comparison methods, details of which can be found in **Table1 of the uploaded PDF file**. In the monolingual test set of the M4 dataset, we achieve the top score, surpassing all other participating schemes. In the multilingual test set, we also secured the fifth place.
> ***
> ## As a work towards detecting AI-generated texts, more analysis of the features of texts should be provided.
> In Section 4.3 of the submission, we provide a visualization analysis of learned text embeddings. Through the UMAP dimensionality reduction visualization, it can be seen that text embeddings of different types have successfully clustered together, which demonstrates the effectiveness of our method.
> ***
> ## I am wondering if this framework would do a good job in authorship attribution?
> We have supplemented the performance of our proposed method on the task of authorship attribution detection on TuringBench dataset, **the results can be found at the top-level Author Rebuttal**. The experimental results indicate that our approach achieves state-of-the-art performance, demonstrating its ability to accomplish this task.
> ***
> ## For the classification approach in ablation studies, is the classification head trained with your own data or initialized in other ways?
> Our classification head is trained with the training data and then tested.
> ***
> ## Could you please provide some explanation on the advantage of multi-level contrastive loss over pcl?
> Please see more explanation in the comment below your review block.
> ***
> ## The authors mention they do not explore the robustness of the proposed method. The validation could be conducted under different attacks like paraphrasing, editing, prompting.
> Thanks for your suggestions. In this paper, our focus is on proposing a general method for AI-generated text detection. As introduced in Section 4.1 of the paper, the M4 dataset includes testing data that has undergone paraphrasing attack, which can be used to verify the model's ability to resist attacks. We have also conducted additional experiments regarding attack robustness which can be found at **the top-level Author Rebuttal and the uploaded PDF file**, and the experimental results demonstrate that our method possesses strong resilience against attacks.

---

> > ### Comment · Reviewer_W7ZQ · 2024-08-11
> > **Response**
> >
> > Thank you for the detailed explanation and additional experiments. Here are some after-rebuttal questions:
> > * Q1
> >
> > My major concern is that this work focuses too much on improving binary classification performance instead of providing insights about synthetic text, which deviates the research topic -- do you think SCL is targeted at detecting synthetic data even if it is a commonly used baseline (even a strong baseline) in this area?
> >
> > Many existing works about AI-generated text detection discuss the differences between AI-generated and human texts, which deepens communities' understanding about AI-generated texts. For example, DetectGPT finds the difference in log probability between two categories. GPT-Sentinal traces integrated gradient to tokens to decide which tokens contribute to the classification. CoCo discovers the coherence structure difference in two categories -- these are examples from the related work introduced in this paper.
> >
> > I agree that the visualization results prove the effectiveness of the learned encoder. However, I argue that this is a success in classification (I think SCL is also able to cluster the data embedding well) but not in explaining the intrinsic difference between AI-generated and human texts -- the research problem in this paper.
> >
> > This is a methodologically inspiring work. I lean to reject this paper for the weak relation to the problem it claims to solve based on the above-mentioned reason. I am willing to increase my score if the authors address my concern.
> >
> > * Q2
> >
> > I have a question about the authorship attribution experiment. If I do not misunderstand, the author indicates the result is in Table 2 in the attached PDF, which is the same (except for the addition of Binoculars) in Table 1 in the original paper. I assume the results in table 1 in the original paper is for binary classification? But authorship attribution is a multi-class classification problem and I did not expect the same results between the two tables. Please tell me if I am wrong about anything.

---

> ### Author Response · Authors · 2024-08-07
> **Answer for the comment "Could you please provide some explanation on the advantage of multi-level contrastive loss over pcl?"**
>
> Suppose we liken our task to appreciating paintings, needing to categorize them into "Impressionist" and "non-Impressionist" genres. If we focus solely on these two broad categories, we might overlook the stylistic differences between works by Monet and Renoir within the Impressionist category. Fine-grained clustering, however, is akin to not only distinguishing between Impressionist and non-Impressionist works but also to further differentiating the styles of various artists within the Impressionist genre itself. When we can differentiate between Monet's and Renoir's works, it means we have gained a deeper understanding of Impressionist pieces, thus making us more adept at distinguishing between the "Impressionist" and "non-Impressionist" categories. This more nuanced classification aids in better understanding and appreciating the artwork, enhancing the precision of our discernment. Similarly, in AI-generated text detection, incorporating multi-level contrastive learning allows the model to more precisely identify and differentiate texts from various sources.
>
> Our ablation study, as shown in Table 3 within the paper, also confirms the multi-level contrastive learning outperforms plain contrastive learning.

---

> ### Author Response · Authors · 2024-08-12
> **(Q1-part1) Response to the after-rebuttal questions raised by Reviewer W7ZQ.**
>
> We sincerely appreciate your acknowledgment of our proposed method as an inspiring work and for the commendable performance of our framework. This recognition serves as an encouragement in our research endeavors. Next, we will address your after-rebuttal questions in detail.
> ## Q1
> Thank you for your thoughtful and detailed feedback. We appreciate the points you've raised and would like to address your concerns directly from the following perspectives.
> * **The research objective and insights of our work.**
>     1. **Performance vs. Insights:** As stated in the Abstract and Introduction of the paper, our insight is that the key to accomplishing AI-generated text detection resides in distinguishing the multi-level writing styles of different "authors", rather than just modeling this task as a binary classification problem. Therefore, based on this key insight, we design the DeTeCtive framework. Specifically, we define multi-level positive and negative sample pairs according to the affinity relations between different language models and derive a multi-level contrastive learning loss function. Eventually, we accomplish the learning of multi-level features through contrastive learning. Moreover, under the DeTective framework, we do not focus on improving binary classification performance. Instead, within our framework, the model's learning objective is to capture multi-level features among different large language models. The principal objective function is the multi-level contrastive learning loss we designed, rather than a simple binary cross-entropy loss. Furthermore, our classification method is based on the K-Nearest Neighbors (KNN) clustering of embeddings. Therefore, the improvement in model performance results from successful learning of multi-level features rather than relying on the classification head. Hence, both the initial intent of framework design and the way of model learning reflect our insights. We believe that the end-to-end method we proposed, which outperforms baselines (e.g., SCL/PCL) and all the comparison methods (e.g., DetectGPT, GPT-Sentinal, CoCo) by a significant margin, is not just a trivial improvement. The substantial performance gains suggest that our method is capturing internal patterns and nuances that are specific to AI-generated text. These internal insights could offer valuable perspectives on how machines differentiate between human and AI-generated content effectively.
>     2. **Human-Friendly Insights vs. Machine Efficiency:** As elaborated above, our understanding and insights of this task are reflected in the design of the method and framework. The visualization of learned text features also reveals that features at different levels (e.g., by model famliy, by individual model) could cluster well, which is something typical supervised contrastive learning (SCL) cannot achieve due to the lack of multi-level relationship constraints. We believe that the insights upheld by our paper, namely, that the key to accomplishing AI-generated text detection task resides in distinguishing the multi-level writing styles of different "authors", is intuitive and human-friendly. The effectiveness of our approach has also been proven through comprehensive experiments and the results are also consistent with our insights.
> * **Weaknesses of existing works.**
>     1. **Unsatisfactory generalizability:** The out-of-distribution (OOD) detection results shown in Table2 of our original submission, the performance of existing methods on OOD data is quite poor. This suggests that the handcrafted features and their insights discovered are challenging to generalize to OOD data, significantly hindering the practical application of the algorithm. Considering the rapid development of language models, this presents a problem.
>     2. **Performance bottlenecks and lack of comprehensive evaluations:** Compared to our work, existing solutions lack a comprehensive evaluation on various benchmarks, testing scenarios and other applications. The experimental results of existing methods on multiple benchmarks are not ideal, indicating that there are performance bottlenecks. However, our method surpasses the existing state-of-the-art solutions by a large margin on each individual dataset.

---

> > ### Comment · Reviewer_W7ZQ · 2024-08-13
> > **Follow-up Response**
> >
> > I appreciate the detailed response from the authors. I totally understand the claimed novelty of this work. But I do not think the authors understand my point. As mentioned in the comments before, if this work is targeted at detecting AI-generated text, analysis of sequence patterns or special tokens should be provided to help the community gain insights about the AI-generated texts. Otherwise, I think this work is also applicable to and should be evaluated on datasets with similar label structures. I will increase my score a bit based on the additional experiments but I encourage the authors to conduct the analysis mentioned above.

---

> > > ### Author Response · Authors · 2024-08-14
> > >
> > > Dear Reviewer,
> > >
> > > We would like to express our sincere gratitude for your response and acknowledgment of our additional experiments. The issue you have raised gives us a new perspective and a direction for reflection. Upon careful consideration, we think that the issue you pointed out could serve as a promising direction for future research.  Once again, we appreciate your valuable comments.
> > >
> > > Authors

---

> ### Author Response · Authors · 2024-08-12
> **(Q1-part2) Response to the after-rebuttal questions raised by Reviewer W7ZQ.**
>
> * **The motivation and contributions of our research work.**
>     1. **Motivation:** In response to the aforementioned weaknesses of existing works, the motivation of our study is to propose an AI-generated text detection algorithm that can be widely applied to various language models and scenarios. It could effectively adapt to newly released large language models and other unseen domains, demonstrating good generalization and robustness.
>     2. **Contributions:** Our work contributes to the field of AI-generated text detection by proposing a novel method that can effectively detect AI-generated text. Based on our framework, we introduce Training-Free Incremental Adaptation (TFIA), a scheme to enhance the model's out-of-distribution (OOD) detection capabilities. We validate our method on multiple benchmarks, where it consistently outperforms existing solutions, especially in terms of OOD detection performance. The visualization of text embeddings also validates that our model captures multi-level text features, which aligns perfectly with our insights. Additionally, we supplement experimental results on attack robustness and authorship attribution detection, further demonstrating our method's superior generalization and robustness surpassing existing solutions. We believe such a novel framework, paired with comprehensive experimental evaluations and state-of-the-art performance, contributes to the community, fostering the application of our algorithm in real-world scenarios.
>
> In conclusion, we believe that our approach offers a complementary perspective that enhances the performance of AI-generated text detection and also aligns well with our key insights. Moreover, we have also conducted corresponding analysis and explorations based on our method, including extensive experiments under various testing scenarios (e.g., multiple benchmarks, OOD detection, authorship attribution detection, attack robustness) and the Training-Free Incremental Adaptation (TFIA) capabilities. These solid experimental results and findings are contributions of our work to the research community. Finally, we are open to incorporate further discussion on interpretability of our method into the final version.

---

> ### Author Response · Authors · 2024-08-12
> **(Q2) Response to the after-rebuttal questions raised by Reviewer W7ZQ.**
>
> ## Q2
> Please allow me to clarify your misconception about our supplementary results for the  task of **authorship attribution detection**.  These corresponding experimental results are detailed in the section titled ***"Supplementary experiments on other applications. (Reviewer yy53, Reviewer W7ZQ)"***, which can be found at the top-level block of ***Author Rebuttal by Authors***. While Table2 of the attched PDF file shows the results of the additional baseline schemes on several existing datasets on the task of AI-generated detection, so the results are consistent with that in Table1 of the original submission. Due to the length constraint of the uploaded PDF file, we only list the additional results of authorship attribution detection in the rebuttal reply. We commit to update these results in the final version afterward.
> ***
> Thank you again for your comments. We sincerely hope that our responses can address your questions.

---

> ### Author Response · Authors · 2024-08-13
>
> Dear reviewer,
>
> We kindly request your feedback on whether our response has addressed your questions. If you have any remaining questions or concerns, we are happy to address them. Thank you for your time and consideration!

---

> > ### Comment · Area_Chair_8Usx · 2024-08-13
> >
> > Dear Reviewer W7ZQ,
> >
> > Thanks again for helping review this paper! Since we are approaching the end of the author-reviewer discussion period, would you please check this author response regarding your concerns? We really appreciate it!
> >
> > Best,
> > AC

---

### Official Review · Reviewer_yy53 · 2024-07-13

**Soundness:** 3
**Presentation:** 3
**Contribution:** 2
**Rating:** 7
**Confidence:** 3

**Summary:**

The paper discusses the challenges of current AI-generated text detection methods, which often suffer from performance issues and poor adaptability to new data and models. The authors introduce a new framework called DeTeCtive, which uses multi-level contrastive learning to distinguish different writing styles rather than just classifying text as human-written or AI-generated. This approach improves the effectiveness of various text encoders, achieving top results across multiple benchmarks, particularly in out-of-distribution scenarios.

DeTeCtive's framework includes a dense information retrieval pipeline and a Training-Free Incremental Adaptation (TFIA) mechanism, enhancing performance without extra training when encountering new data. The method fine-tunes text encoders using a new multi-task auxiliary, multi-level contrastive learning loss to capture detailed features of different writing styles.

Extensive experiments show that DeTeCtive surpasses existing methods in detecting AI-generated text and excels at handling data from unseen models and domains. The paper also emphasizes the method's compatibility with various text encoders and its strong performance in diverse scenarios, making a significant contribution to the field of AI-generated text detection and promoting the safe use of large language models.

**Strengths:**

The use of multi-level contrastive learning to distinguish writing styles is a novel method that goes beyond traditional binary classification. This allows for a more nuanced detection of AI-generated text, improving the overall detection performance.
The proposed method consistently outperforms existing techniques across multiple benchmarks, establishing new state-of-the-art results. This indicates the effectiveness of the multi-level contrastive learning framework.

**Weaknesses:**

While the paper shows strong performance on the chosen benchmarks, it remains unclear how well the model would perform on other datasets or in different application contexts. This limits the generalizability of the findings to some extent.

**Questions:**

How does the quality and diversity of the training data affect the performance of DeTeCtive? Are there specific datasets that are more beneficial for training the model?

**Limitations:**

The focus on distinguishing writing styles might overlook content-based cues that could also indicate AI-generated text. This might lead to missed detections if an AI successfully mimics human writing style while generating misleading content.

---

> ### Author Rebuttal · Authors · 2024-08-07
>
> *We would like to thank Reviewer yy53 for the valuable feedback. The comments and suggestions have greatly helped us in improving the quality of our work. Please see below for our responses to your comments.*
> ***
> ## The paper's strong performance may not generalize to other datasets or applications.
> We appreciate your acknowledgment towards our proposed framework and experimental results. In the submission, we conduct evaluations on three widely-used datasets for AI-generated text detection. These include a variety of testing scenarios, such as: multi-model, multi-domain, multi-language, text-paraphrase attack, and out-of-distribution (OOD) detection, among others. The experimental results demonstrate that our method achieves state-of-the-art performance in these various scenarios. Furthermore, we supplement our experiments on **attack robustness** and **authorship attribution detection**, and we also add performance comparisons with **more baseline methods** on existing datasets. Please refer to **the tabels in the top-level Author Rebuttal** and **the uploaded PDF file** for details. Additionally, we supplement **OOD evaluation** based on the "Unseen Domains & Unseen Models" dataset from the paper "MAGE: Machine-generated Text Detection in the Wild" presented at ACL 2024. The testing data is entirely derived from GPT-4, with the corresponding results presented in the table below. The results also indicate that our method performs better than other solutions. These additional experimental results will be incorporated into the final version of our paper.
>
> To sum up, our approach exhibits state-of-the-art performance across various datasets, testing scenarios, and applications.
> | Methods | HumanRec | MachineRec | AvgRec | F1 |
> |---|---|---|---|---|
> | Unseen Domains & Unseen Model |  |  |  |  |
> | FastText | 71.78 | 68.88 | 70.33 | 70.21 |
> | GLTR | 16.79 | 98.63 | 57.71 | 28.40 |
> | Longformer | 52.50 | 99.14 | 75.82 | 68.45 |
> | Longformer† | 88.78† | 84.12† | 86.54† | 86.42 |
> | **DeTeCtive** | **75.46** | **97.88** | **86.67** | **84.93** |
>
> (Longformer† indicates that Longformer uses data from the testing set to determine the detection threshold.)
> ***
> ## How does the quality and diversity of the training data affect the performance of DeTeCtive?
> In the experiments, we use publicly available datasets and strictly follow the divisions of the training and testing sets without introducing any additional data to assist model training.
> Moreover, we believe that increasing data diversity and quality can effectively enhance the performance of our method. Under the proposed multi-level contrastive learning framework, diverse data compose more positive and negative sample pairs, enabling the model to learn more fine-grained writing-style features and relationships, thereby improving the model's performance. We believe that scaling up the data volume within our proposed framework holds promise as a future research direction.
> ***
> ## Are there specific datasets that are more beneficial for training the model?
> Within our proposed framework, the process is streamlined, requiring only the texts and labeling of different large language models. Following this, our method facilitates the construction of multi-level contrastive learning sample pairs. Consequently, there is no necessity for specific datasets to train our model.
> ***
> ## The focus on distinguishing writing styles might overlook content-based cues that could also indicate AI-generated text. This might lead to missed detections if an AI successfully mimics human writing style while generating misleading content.
> We appreciate you pointing out the potential limitation of our method. We will address this issue from the following two aspects, hoping to alleviate your concerns:
> 1.  As our approach is based on fine-tuning pre-trained text encoders, which already possess the capability of content-based semantic understanding (such as BERT, SimCSE, etc.), we believe that the model can understand the content and semantics of the text itself.
> 2. Further, we conduct paraphrasing-attack experiments for validation. Specifically, OUTFOX and DIPPER paraphrase texts generated by large language models to mimic human-written texts, employing this as a paraphrasing attack strategy. We train our model on the training set provided by OUTFOX and test under the following three scenarios: Non-attacked, OUTFOX paraphrasing attack, and DIPPER paraphrasing attack. We also compare with several baseline methods, and the results are as follows:
> | Attacker | Detector | HumanRec | MachineRec | AvgRec | F1 |
> |---|---|---|---|---|---|
> | Non-attacked | RoBERTa-base | 93.8 | 92.2 | 93.0 | 92.9 |
> |  | RoBERTa-large | 91.6 | 90.0 | 90.8 | 90.7 |
> |  | HC3 detector | 79.2 | 70.6 | 74.9 | 73.8 |
> |  | OUTFOX | 99.0 | 94.0 | 96.5 | 96.4 |
> |  | **DeTeCtive** | **98.2** | **100.0** | **99.1** | **99.1** |
> | DIPPER | RoBERTa-base | 93.8 | 89.2 | 91.5 | 91.3 |
> |  | RoBERTa-large | 91.6 | 97.0 | 94.3 | 94.4 |
> |  | HC3 detector | 79.2 | 3.4 | 41.3 | 5.5 |
> |  | OUTFOX | 98.6 | 66.2 | 82.4 | 79.0 |
> |  | **DeTeCtive** | **97.4** | **97.9** | **97.7** | **97.5** |
> | OUTFOX | RoBERTa-base | 93.8 | 69.2 | 81.5 | 78.9 |
> |  | RoBERTa-large | 91.6 | 56.2 | 73.9 | 68.3 |
> |  | HC3 detector | 79.2 | 0.4 | 39.8 | 0.7 |
> |  | OUTFOX | 98.8 | 24.8 | 61.8 | 39.4 |
> |  | **DeTeCtive** | **95.4** | **98.6** | **97.0** | **96.9** |
>
> The results indicate that even when AI-generated texts are subjected to paraphrasing attack to mimic human writings, DeTeCtive experiences only a minimal performance decline compared to the non-attacked scenario and remains effective in detecting AI-generated text. In contrast, other solutions we compared all exhibit significant performance degradation. Therefore, we believe that our framework demonstrates good robustness against mimicry of human writing style.

---

### Author Rebuttal · Authors · 2024-08-07

We sincerely thank all the reviewers for their reviews and constructive feedback. Taking into account each concern and question posed by the reviewers, we have given thorough responses within our rebuttal. It is our hope that our responses will be kindly considered during the evaluation of our submission by the reviewers and the AC. Here, we summarize the primary concerns raised by the reviewers and provide a consolidated response.
***
## The elaboration on the effectiveness and timeliness of the comparison methods. (Reviewer W7ZQ, Reviewer Tv1c)
The benchmarks and comparison methods used in our submission are state-of-the-art research works in the field of AI-generated text detection. Here, we summarize all the comparison methods and benchmarks information for your reference. (Note, symbol ("*") denotes supplementary methods during the rebuttal phase.)
| Method | Paper title | Conference |
|---|---|---|
| FastText | Bag of Tricks for Efficient Text Classification | EACL 2017 |
|  GLTR | GLTR: Statistical Detection and Visualization of Generated Text | ACL 2019 |
| SCL | Supervised Contrastive Learning for Pre-trained Language Model Fine-tuning | ICLR 2021 |
| DetectGPT | DetectGPT: Zero-Shot Machine-Generated Text Detection using Probability Curvature | ICML 2023 |
|  DIPPER | Paraphrasing evades detectors of AI-generated text, but retrieval is an effective defense | NeurIPS 2023 |
| T5-Sentinel | Token Prediction as Implicit Classification to Identify LLM-Generated Text | EMNLP 2023 |
| OUTFOX | OUTFOX: LLM-Generated Essay Detection Through In-Context Learning with Adversarially Generated Examples |  AAAI 2024 |
| Longformer (MAGE) | MAGE: Machine-generated Text Detection in the Wild | ACL 2024 |
| GPT-who* | GPT-who: An Information Density-based Machine-Generated Text Detector | NAACL 2024 |
| Binoculars* | Spotting LLMs With Binoculars: Zero-Shot Detection of Machine-Generated Text | ICML 2024 |

| Benchmark        | Conference |
| --------------- | ---------- |
| TuringBench     | EMNLP 2021 |
| M4              | NAACL 2024 |
| Deepfake (MAGE) | ACL 2024   |
| OUTFOX          | AAAI 2024  |
***
## The study of robustness of our methodology against attacks. (Reviewer W7ZQ, Reviewer Tv1c)
As stated in **Section 4.1 of our submission**, testing data of the M4 dataset has undergone paraphrase attacks using the OUTFOX method. We believe evaluations on this dataset can serve as a reliable indication of our model's robustness against such attacks. Further, as acknowledged in the **Limitation and Future Work section of our paper**, our focus primarily lies in proposing a detection algorithm rather than an extensive study into defensive robustness, whereas the latter being a specialized realm of study. Nevertheless, based on the valuable feedback from the reviewers, we supplement our research with experiments on attack robustness using the OUTFOX dataset, and results are as follows.
| Attacker | Detector | HumanRec | MachineRec | AvgRec | F1 |
|---|---|---|---|---|---|
| Non-attacked | RoBERTa-base | 93.8 | 92.2 | 93.0 | 92.9 |
|  | RoBERTa-large | 91.6 | 90.0 | 90.8 | 90.7 |
|  | HC3 detector | 79.2 | 70.6 | 74.9 | 73.8 |
|  | OUTFOX | 99.0 | 94.0 | 96.5 | 96.4 |
|  | **DeTeCtive** | **98.2** | **100.0** | **99.1** | **99.1** |
| DIPPER | RoBERTa-base | 93.8 | 89.2 | 91.5 | 91.3 |
|  | RoBERTa-large | 91.6 | 97.0 | 94.3 | 94.4 |
|  | HC3 detector | 79.2 | 3.4 | 41.3 | 5.5 |
|  | OUTFOX | 98.6 | 66.2 | 82.4 | 79.0 |
|  | **DeTeCtive** | **97.4** | **97.9** | **97.7** | **97.5** |
| OUTFOX | RoBERTa-base | 93.8 | 69.2 | 81.5 | 78.9 |
|  | RoBERTa-large | 91.6 | 56.2 | 73.9 | 68.3 |
|  | HC3 detector | 79.2 | 0.4 | 39.8 | 0.7 |
|  | OUTFOX | 98.8 | 24.8 | 61.8 | 39.4 |
|  | **DeTeCtive** | **95.4** | **98.6** | **97.0** | **96.9** |

The analysis is as follows:

Through training with our proposed multi-level contrastive learning framework, we can discern fine-grained features of AI-generated and human-written texts. Furthermore, our usage of the K-Nearest Neighbours (KNN) algorithm for classification offers our approach with a level of fault tolerance. Thus, minor disturbances prompted by certain attacks do not engender significant feature drift. Consequently, our method remains both effective and robust in detection.
***
## Supplementary experiments on other applications. (Reviewer yy53, Reviewer W7ZQ)
Taking into account the valuable feedback from the reviewers, we conduct **authorship attribution detection** on TuringBench dataset. The results confirm that our method also demonstrates state-of-the-art performance on this task.
| Model | Precision | Recall | F1 | Accuracy |
|---|---|---|---|---|
| Random Forest | 58.93 | 60.53 | 58.47 | 61.47 |
| SVM (3-grams) | 71.24 | 72.23 | 71.49 | 72.99 |
| WriteprintsRFC | 45.78 | 48.51 | 46.51 | 49.43 |
| OpenAI detector | 78.10 | 78.12 | 77.41 | 78.73 |
| Syntax-CNN | 65.20 | 65.44 | 64.80 | 66.13 |
| N-gram CNN | 69.09 | 68.32 | 66.65 | 69.14 |
| N-gram LSTM-LSTM | 66.94 | 68.24 | 66.46 | 68.98 |
| BertAA | 77.96 | 77.50 | 77.58 | 78.12 |
| BERT-Multinomial | 80.31 | 80.21 | 79.96 | 80.78 |
| RoBERTa-Multinomial | 82.14 | 81.26 | 81.07 | 81.73 |
| **DeTeCtive** | **84.04** | **82.59** | **83.05** | **82.75** |
***
Finally, we address each reviewer's comments in detail below their reviews. **The attached PDF file** further includes comprehensive comparisons of more baseline methods on three existing datasets **(see Table 2)**, including the **Binoculars** method. Additionally, we have added comparisons with **GPTZero** and **GPTWho** methods on the Deepfake dataset **(see Table 3)**. Furthermore, it includes a performance comparison with a recent competition (more than 100 participating teams) based on the M4 dataset **(see Table 1)**. Please kindly check out them. We will include all additional experiment results **in our final version**. Thank you and we hope that our submission can be fully discussed in the next stage.

---

### Decision · Program_Chairs · 2024-09-25

**Decision:**

Accept (poster)

**Comment:**

This paper proposes to detect AI-generated text by formulating the problem as distinguishing writing styles of different authors. The proposed framework uses distinguishing writing styles, and leverages den information retrieval and training-free incremental adaptation mechanism to deal with new data and models. Experiment demonstrates its state-of-the-art performance, notably for OOD settings. Reviewers appreciate the contribution of this work because of its innovation in tackling OOD LM-generated text detection and its solid experimental comparisons. Rebuttal and discussions further addressed reviewer's concerns, and the consensus emerged. Please include the extra results and analysis on watermarking and authorship attribution in the final version.